# Clinical Characteristics and Outcomes of Persistent Staphylococcal Bacteremia in a Tertiary Care Hospital

**DOI:** 10.3390/antibiotics12030454

**Published:** 2023-02-24

**Authors:** Shiori Kitaya, Hajime Kanamori, Yukio Katori, Koichi Tokuda

**Affiliations:** 1Department of Infectious Diseases, Internal Medicine, Tohoku University Graduate School of Medicine, Sendai 980-8574, Japan; 2Department of Otolaryngology, Head and Neck Surgery, Tohoku University Graduate School of Medicine, Sendai 980-8574, Japan

**Keywords:** persistent bacteremia, *Staphylococcus aureus*, coagulase-negative staphylococci, methicillin resistance

## Abstract

Clinical outcomes of persistent staphylococcal bacteremia vary depending on the causative organism. This secondary data analysis study compared the clinical characteristics of persistent *Staphylococcus aureus* (*S. aureus*)- and coagulase-negative staphylococci (CoNS)-caused bacteremia, focusing on the methicillin-resistant status. This study used data collected from patients who underwent blood cultures between January 2012 and December 2021 at Tohoku University Hospital, Japan. Patients with persistent staphylococcal bacteremia were divided into groups based on the pathogen and methicillin-resistant status, and their characteristics were analyzed. The primary outcomes were early (30-day), late (30–90 days), and 90-day mortality rates. The early, late, and 90-day mortality rates were similar between the persistent CoNS and *S. aureus* bacteremia groups. Patients with persistent methicillin-resistant *S. aureus* (MRSA) bacteremia tended to have higher early, late, and 90-day mortality rates than those with persistent methicillin-susceptible *S. aureus* bacteremia (not statistically significant). No differences were observed between the methicillin-resistant and-susceptible CoNS groups. In patients with persistent CoNS bacteremia, mortality tended to increase, especially in debilitated or immunocompromised patients with distant metastases, underscoring the importance of infection source control. Mortality tended to be high in patients with persistent MRSA bacteremia, especially when persistent bacteremia clearance was not confirmed, illustrating the need for careful therapeutic management.

## 1. Introduction

Gram-positive cocci (GPC) are common agents in short-term and persistent bacteremia (PB), as demonstrated in numerous reports [1,2,3]. In our previous study, 13.2% of all patients with positive blood culture (BC) had PB, and GPC was the causative organism in 53.1% [4].

*Staphylococcus aureus* (*S. aureus*) is the most common cause of PB [5]. Between 8% and 39% of *S. aureus* bacteremia patients progress to PB, particularly endovascular infections [6,7]. In multivariate analysis, independent risk factors for persistent *S. aureus* bacteremia included community-onset bacteremia, bone and joint infection, infective endocarditis (IE), central venous catheter-related infection, metastatic infection, and methicillin resistance [7,8]. Among coagulase-negative staphylococci (CoNS), the incidence of PB is reported to be between 25–40%, with *Staphylococcus epidermidis* (*S. epidermidis*) being the most common cause of PB [3,9,10,11]. Atypical outbreaks of persistent CoNS bacteremia have been reported in neonatal intensive care units (NICUs) [11,12,13] despite aggressive antibiotic therapy, with biofilm formation and central venous catheterization being identified as risk factors for the development of persistent CoNS bacteremia [3,9,11].

Recently, there has been a growing focus on PB caused by antimicrobial-resistant bacteria. Risk factors for persistent methicillin-resistant *S. aureus* (MRSA) bacteremia include liver cirrhosis, endocarditis, bone and joint infections, metastatic infection, septic shock, complicated bacteremia, C-reactive protein level ≥ 10 mg/dL, diminished susceptibility to vancomycin, vancomycin heteroresistance, accessory gene regulator dysfunction, and low-level in vitro resistance to thrombin-induced platelet microbicidal proteins [14,15,16]. Metastatic infection, congestive heart failure, and increasing vancomycin minimal inhibitory concentrations (MICs) are independent predictors of 30-day mortality in MRSA bacteremia [17]. A report has shown that 98% of PB-causing bacteria in CoNS are methicillin-resistant, and the success rate of antimicrobial therapy significantly increases with the removal of a catheter, owing to reducing the inhibition of activity of glycopeptide antibiotics to the microorganisms in biofilms [18]. 

Recent studies on persistent *S. aureus* and CoNS bacteremia have focused on the risk factors for PB caused by each species. However, few studies have compared the clinical characteristics of PB caused by these species. Additionally, most CoNS studies have primarily focused on PB among infants in the NICUs, and information on persistent CoNS bacteremia in adults is lacking. Furthermore, few studies have compared the outcomes of methicillin resistance in persistent *S. aureus* and CoNS bacteremia. Therefore, this study aimed to clarify the differences in the clinical characteristics of persistent *S. aureus* and CoNS bacteremia in terms of causative species and methicillin resistance.

## 2. Results

### 2.1. Clinical Characteristics of Persistent S. aureus and CoNS Bacteremia 

Significant differences were observed in vital signs and laboratory markers between the persistent *S. aureus* and CoNS bacteremia groups (Table 1), with body temperature, white blood cell count, neutrophil count, and C-reactive protein being significantly higher in the former group (*p* = 0.009, *p* = 0.022, *p* = 0.006, and *p* < 0.001, respectively). Additionally, the ratios of hematological malignancy, neutropenia, and immunosuppression were significantly lower in the persistent *S. aureus* bacteremia group than in the persistent CoNS bacteremia group (odds ratio (OR) = 0.1, *p* = 0.001; OR = 0.1, *p* = 0.032; and OR = 0.4, *p* = 0.025, respectively). The persistent *S. aureus* bacteremia group had significantly lower rates of intravascular device insertion and catheter-related bloodstream infection (CRBSI) than the persistent CoNS bacteremia group (OR = 0.2, *p* < 0.001 and OR = 0.1, *p* < 0.001, respectively). However, the persistent *S. aureus* bacteremia group had significantly higher ratios of abscesses, IE, endovascular device infections, septic embolism, and pyogenic spondylitis than the persistent CoNS bacteremia group (OR = 11.4, *p* = 0.003; OR = 5.2, *p* = 0.018; OR = 8.9, *p* = 0.017; OR = 7.3, *p* = 0.030; and OR = 7.3, *p* = 0.030, respectively). The rates of admission to the intensive care unit (ICU) and the duration of stay in the ICU were significantly lower in the persistent *S. aureus* bacteremia group than in the persistent CoNS bacteremia group (OR = 0.5; *p* = 0.046 and *p* = 0.034, respectively). There was no significant difference in mortality between the persistent *S. aureus* and CoNS bacteremia groups. 

### 2.2. Clinical Characteristics of Persistent S. aureus and CoNS Bacteremia in Terms of Methicillin Resistance

The clinical characteristics of persistent *S. aureus* and CoNS bacteremia in terms of methicillin resistance are described in detail in Table 2. There were no significant differences between persistent MRSA and methicillin-susceptible *S. aureus* (MSSA) bacteremia or between persistent MRCoNS and MSCoNS bacteremia regarding vital signs, laboratory markers, site of infection, and the presence or duration of ICU, high care unit, and cardiac care unit stay. The duration of *S. aureus* bacteremia was significantly longer in the persistent MRSA bacteremia group than in the persistent MSSA bacteremia group (*p* < 0.001). 

Although there were no statistically significant differences, the persistent MRSA bacteremia group tended to have higher mortality rates in the early stages (within 30 days) and 90 days than the persistent MSSA bacteremia group. However, clinical characteristics, including mortality, in the CoNS bacteremia group did not differ significantly based on the presence or absence of methicillin resistance. The rate of PB non-clearance was higher in the persistent MRSA bacteremia group (10/43 cases, 23.3%) than in the persistent MSSA bacteremia group (4/49 cases, 8.2%). Non-clearance rates in persistent CoNS bacteremia were similar in the MRCoNS (22/70 cases, 31.4 %) and MSCoNS (5/18 cases, 27.8 %) groups. The early (30-day), late (30–90 days), and 90-day mortality rates for persistent MRSA bacteremia tended to be higher in the non-clearance PB group than in the clearance PB group, although the differences were not statistically significant (2/10 (20%) vs. 2/33 (6.1%), 1/10 (10%) vs. 0/33 (0%), and 3/10 (30%) vs. 2/33 (6.1%) patients, respectively). Conversely, there were no discernible variation in early (30-day), late (30–90 days), and 90-day mortality rates among patients with persistent MSSA bacteremia, irrespective of the presence or absence of PB clearance (non-clearance: 0/4 (0%) vs. 1/45 (2.2%), 0/4 (0%) vs. 1/45 (2.2%), and 0/4 (0%) vs. 2/45 (4.4%) patients, respectively). The persistent CoNS bacteremia group showed no significant differences in the early (30-day), late (30–90 days), and 90-day mortality rates, regardless of the presence or absence of PB clearance in the methicillin-resistant and non-resistant subgroups (methicillin-resistant non-clearance; clearance; 2/22 (9.1%) vs. 0/48 (0%), 0/22 (0%) vs. 3/48 (6.3%), and 2/22 (9.1%) vs. 3/48 (6.3%) patients, respectively; methicillin-susceptible non-clearance; clearance; 0/5 (0%) vs. 0/13 (0%), 0/5 (0%) vs. 1/13 (7.7%), and 0/5 (0%) vs. 1/13 (7.7%), respectively).

### 2.3. Reasons for Non-Clearance of Persistent Staphylococcal Bacteremia

Figure 1 shows the reasons for the non-clearance of persistent staphylococcal bacteremia. The most common cause of failure to confirm the clearance of persistent *S. aureus* bacteremia was inadequate treatment, which was significantly more common than in the persistent CoNS bacteremia group (*p* = 0.001). The following most common causes of failure to confirm clearance of persistent *S. aureus* bacteremia were unknown (*p* = 0.024), lack of confirmation of negative BCs due to best supportive care (BSC) policy, death of a patient from a disease other than an infectious disease, and unknown focus of infection. However, in many patients with persistent CoNS bacteremia, follow-up BCs (FUBCs) were not performed because of appropriate antimicrobial therapy and good source control in CRBSIs, which were significantly more common than those in the persistent *S. aureus* bacteremia group (*p* = 0.001). The next most common causes of failure to confirm clearance of persistent CoNS bacteremia were patients in whom FUBCs were not performed due to BSC policy or the death of a patient from a disease other than an infectious disease, lack of confirmation of negative BCs due to improvement in general condition, contamination, and unknown.

## 3. Discussion

### 3.1. Differences in Clinical Characteristics between Persistent S. aureus and CoNS Bacteremia 

Our hospital, a university facility with many departments, actively treats patients with hematological malignancies. Because of the administration of immunosuppressive drugs in combined with anticancer therapy and bone marrow transplantation, such patients are frequently immunocompromised, resulting in a presumed high incidence of neutropenia. In fact, among the persistent CoNS bacteremia patients with hematologic malignancies in this study, 14 of 15 (93.3%) were immunosuppressed, and 7 of the 14 (50.0%) developed neutropenia. These associations revealed that the persistent CoNS bacteremia group had higher rates of hematological malignancy, neutropenia, and immunosuppression than the persistent *S. aureus* bacteremia group. A higher number of patients in the persistent CoNS bacteremia group with hematological malignancies and neutropenia may have contributed to the significantly lower white blood cell and neutrophil counts in the persistent CoNS bacteremia group than in the persistent *S. aureus* bacteremia group. Furthermore, our hospital provides aggressive treatment for children with acute lymphoblastic leukemia and acute myeloid leukemia, which may have contributed to the persistent CoNS bacteremia group being significantly younger than the persistent *S. aureus* bacteremia group. The frequent administration of drugs in patients with hematologic malignancies often necessitates the implantation of intravascular devices to facilitate long-term systemic treatment [19]. In the present study, all 15 patients (100%) with persistent CoNS bacteremia and hematologic malignancies had implanted devices. Biofilm formation facilitates the adherence and colonization of central venous catheters (CVCs) and can lead to CRBSIs [20]. The colonization of surfaces and formation of biofilms by CoNS bacteria have long been considered their primary virulence factors, with evidence indicating that biofilm formation occurs in nearly all CoNS bacterial infections. However, it is present in approximately 20% of *S. aureus* infections [21,22,23]. Additionally, CoNS species, such as *S. epidermidis* and *Staphylococcus haemolyticus* (*S. haemolyticus*), have been identified as the most common causes of CVC colonization and CRBSIs [23,24,25,26], and our findings are consistent with those of previous reports (Appendix A).

Metastatic infection is a significant complication of *S. aureus* bacteremia, and failure to identify the infection can result in persistent or recurrent bacteremia. Previous studies have shown that the incidence of metastatic infections due to *S. aureus* bacteremia ranges from 13% to 39% [18]. In this study, 50 of the 92 (54.3%) patients had metastatic infections, with abscesses, IE, and endovascular device infections being the most common sites of localization. The higher rate of metastatic infections in this study compared to previous reports may be due to the practice in our hospital, with the Department of Infectious Diseases encouraging physicians to examine metastatic infections in all patients with positive *S. aureus* BCs. Consultation with the Department of Infectious Diseases for episodes of *S. aureus* bacteremia generally leads to a more thorough evaluation, detection of IE and metastatic complications, and improved adherence to treatment guidelines [27,28]. Of the *S. aureus* bacteremia patients complicated by metastatic infection in our hospital, 16 of the 50 (32.0%) patients had good source control of the metastatic infection, such as drainage, and no deaths occurred. However, 34 of the 50 (68.0%) patients had poor source control measures, and 3 of these 34 (8.8%) patients died. Previous research indicated that source control of metastatic infection is associated with a favorable prognosis in *S. aureus* bacteremia [29]. These results suggest that source control of metastatic infections may help improve survival rates in persistent *S. aureus* bacteremia.

In the present study, mortality in the persistent CoNS bacteremia group was comparable to that observed in the persistent *S. aureus* bacteremia group, regardless of the presence or absence of PB clearance. Patients with persistent CoNS bacteremia admitted to our hospital often have a poor general condition and are at a high risk of CRBSIs due to factors such as immunosuppression and neutropenia resulting from treatment for hematologic malignancies, as well as the insertion of intravascular devices for treatment and nutritional support. The persistent CoNS bacteremia deaths (6 patients) had underlying conditions, such as chronic renal insufficiency, after organ transplantation, acute pancreatitis, and ruptured thoracoabdominal aneurysm, and almost all (5 of 6 patients, 83.3%) had indwelling endovascular devices. Previous studies have reported how implantable medical devices and the increasing number of vulnerable patients owing to advances in medicine have allowed CoNS to cause significant infections in humans with higher mortality rates [30,31,32]. The infection focus in these six deaths was CRBSI in five patients and unknown in one patient. Of the five CRBSI patients, three were complicated by metastatic infection, and in all three instances, metastatic infection was not effectively managed at the source. However, no deaths were observed among patients in the persistent CoNS bacteremia group, who had effective source control measures for distant metastases (0%, 0/3 patients). This suggests that the lack of adequate source control of distant metastases may have contributed to increased mortality in patients with persistent CoNS bacteremia in this study. 

### 3.2. Differences in Clinical Characteristics between Persistent S. aureus and CoNS Bacteremia in Terms of Methicillin Resistance

Methicillin-resistant *S. aureus* bacteremia can elicit a more virulent and invasive infectious disease process, accompanied by a paucity of antibiotic treatment options, prolonged hospitalization, and frequent need for intensive care and surgical intervention, which may contribute to a longer duration of bacteremia and increased mortality [33]. Previous research has demonstrated that MRSA bacteremia (with a median duration of 8–9 days for clearance) tends to persist longer than MSSA bacteremia (with a median duration of 3 days) [34], and the mortality rate for MRSA bacteremia is as high as 60%, which is twice that of MSSA bacteremia [35]. This study demonstrated that even in patients with PB, the duration of bacteremia tended to be longer, and the mortality rate was higher when MRSA, rather than MSSA, was the causative organism. 

### 3.3. Causes of Non-Clearance of Persistent Staphylococcal Bacteremia

In the persistent *S. aureus* bacteremia group, the sources of infection in patients in whom PB was not cleared due to inadequate source control were CVC (one patient) and endovascular devices, such as ventricular assist devices, cardiovascular implantable electronic devices, and vascular grafts (one patient each), with half of these patients (two of four patients, 50%) resulting in death. Current guidelines recommend immediate removal of CVCs for CRBSI caused by *S. aureus* [36,37]. In this study, of the CRBSI patients in the persistent *S. aureus* bacteremia group, 38 of 40 (95%) patients with appropriate source control measures, such as CVC removal and replacement, had no deaths, whereas one of the two patients (50%) without appropriate source control measures died. These results suggest the importance of source control measures for persistent CRBSI caused by *S. aureus*.

Failure to remove intravascular devices infected with *S. aureus*, which causes recurrent bacteremia, has been reported to result in morbidity and mortality, including the development of metastatic foci of infection [38,39,40]. In this study, of the 12 patients in the persistent *S. aureus* bacteremia group with endovascular device infection, 2 (16.7%) with appropriate source control measures did not die, and 2 of 10 (20%) patients without appropriate source control measures died. However, the removal of endovascular devices is often highly invasive. It requires large-scale surgery rather than CVC removal, making it more difficult in some patients, depending on their surgical tolerance and general condition. In the present study, adequate source control measures were achieved in 38 of 40 (95%) patients with CRBSI; however, only 2 of 12 (16.7%) patients had endovascular device infections. Guidelines for managing infection in left ventricular assist device (LVAD) recipients state that device replacement is an option. However, it is associated with increased intraoperative risk and increased likelihood of recurrence [41]. Some studies have suggested that aggressive surgical incision, drainage, and debridement of the driveline and LVAD pocket infection should be attempted, whenever possible, before device replacement [42]. A previous review reported that LVAD infections were well-controlled in most patients with local debridement of the exit site in cases of driveline infections [43]. Focusing on the presence or absence of PB clearance in the ten patients with endovascular device infection in whom appropriate source control measures were not performed, seven patients were confirmed to have cleared PB, and only one (14.3%) died. However, 1/3 (33.3%) of patients without confirmed PB clearance died. Although the number of patients was small, these results suggest that source control measures are essential for improving survival in CRBSI and endovascular device infections in patients with persistent *S. aureus* bacteremia and that confirming PB clearance may help improve survival, especially when source control measures are difficult due to systemic conditions or invasive problems in endovascular device infections.

Coagulase-negative staphylococci are generally less pathogenic than *S. aureus*, except for certain strains such as *Staphylococcus lugdunensis* (*S. lugdunensis*) [25]. Additionally, the current guidelines for the management of CRBSI do not recommend aggressive removal of CVCs when the causative organism is CoNS, in contrast to the strong recommendation for *S. aureus*, and the recommended duration of antimicrobial therapy is shorter for CoNS than for *S. aureus* [37]. Fortunately, in this study, there were no deaths among patients with CRBSI who did not undergo FUBC and were not confirmed to have clearance of persistent CoNS bacteremia, owing to the completion of an appropriate course of antimicrobial therapy and/or the implementation of appropriate source control. However, in patients with poor general conditions, including those who are immunosuppressed, persistent CoNS bacteremia can sometimes be fatal, as observed in the current study. Therefore, considering the causative organisms, patient characteristics, and overall condition, it is important to consider the addition of FUBCs and the extension of the treatment period.

## 4. Materials and Methods

### 4.1. Study Design and Setting

For secondary data analyses, we used electronic clinical charts and hospital records from Tohoku University Hospital between January 2012 and December 2021. Data for this study were extracted from the computerized records of the Department of Laboratory Medicine and medical records and databases of the Department of Infectious Diseases, Tohoku University Hospital. The variables investigated included demographic information (age, sex, comorbidities, body mass index, and body temperature), blood test results (levels of serum white blood cells, neutrophils, and C-reactive protein), clinical characteristics (presence and removal of intravascular devices, history of cardiovascular surgery, use of extracorporeal membrane oxygenation, continuous hemodiafiltration, mechanical ventilation, interval between initial BC and FUBC, duration of bacteremia, site of infections, duration of hospitalization, time spent in the ICU, high care unit, or coronary care unit, antibiotic use, performance of source control, and early, late, and 90-day mortality). Intravascular devices include conventional CVCs, peripherally inserted central catheters, tunneled CVCs, and implanted central venous ports. Cardiovascular surgeries include valve replacement, vascular graft replacement, ventricular-assisted device implantation, and cardiac device implantation. Endovascular device infections involve vascular grafts, pacemakers, implantable cardioverter defibrillators, and left ventricular assist devices.

All patients diagnosed with BSI, defined as one or more positive BCs to rule out infection, were eligible for inclusion. The exclusion criteria included PB patients under 18 years of age, presence of polymicrobial PB, and possible contaminants (such as CoNS, *Propionibacterium* spp., and *Corynebacterium* spp.). Microbial data were extracted from the Infectious Disease Department database. The Infectious Diseases Department reviews all patients with persistent staphylococcal bacteremia at our hospital and recommends routine FUBC, echocardiography, and checks for metastatic infections, particularly persistent *S. aureus* bacteremia.

The Human Ethics and Clinical Trial Committee of Tohoku University Hospital approved this study (2019-1-270). The requirement for patient consent was waived because of the retrospective nature of the study.

### 4.2. Definitions and Outcomes 

Definitions of PB and FUBC, determination of the source of PB, duration of bacteremia, PB clearance, source control, neutropenia, immunosuppression, BC collection, adequacy of antimicrobial therapy, and patient selection algorithm were determined in accordance with our previous report [4]. The investigator determined coagulase-negative staphylococci to be contaminants if they did not fulfill the previously validated criteria for a significant isolate [25]. The underlying diseases in the patients for whom FUBC was not performed because of the BSC treatment policy were gastric cancer, interstitial pneumonia, cerebral infarction, and disseminated intravascular coagulation. Non-infectious diseases refer to patients in whose clearance of PB cannot be confirmed due to death from a cause other than an infectious disease, such as interstitial pneumonia or acute pancreatitis. Inappropriate treatment included patients for whom the duration of antimicrobial therapy, selection of antimicrobial drug, or source control measures was inadequate. This study identified the following CoNS strains as the causative agents of PB: *S. epidermidis*, *Staphylococcus capitis*, *Staphylococcus caprae*, *S. haemolyticus*, *Staphylococcus hominis*, *S. lugdunensis*, *Staphylococcus piscifermentans*, and *Staphylococcus warneri*. VITEK MS System (BioMérieux, Métropole de Lyon, France) was used to identify bacterial species of *S. aureus* and CoNS, and a Walk Away-96 Plus system (Siemens Healthcare Diagnostics, Deerfield, IL, USA) was used for susceptibility testing.

In this study, the primary outcome variables were early (30-day), late (30–90 days), and 90-day mortality rates after the initial BC. Early (30-day), late (30–90 days), and 90-day mortalities were defined as deaths due to persistent staphylococcal bacteremia. The MIC results and clinical breakpoints for each bacterium were evaluated according to the Clinical and Laboratory Standards Institute [44]. 

### 4.3. Statistical Analysis

The results of this study are expressed as the median value with a 95% confidence interval (95% CI) or as a proportion of the total number of patients or isolates. For comparisons between the two groups, the Mann–Whitney U test was used to compare the averages of continuous variables, and Fisher’s exact test was used to compare the proportions of categorical variables. To compare differences in the characteristics of PB due to *S. aureus*, *S. epidermidis*, and other CoNS, we analyzed continuous variables using the Kruskal–Wallis test followed by the Steel–Dwass test and categorical variables using Fisher’s exact test. All *p*-values were corrected using Ryan–Holm step-down Bonferroni correction. Analysis was performed using JMP Pro 16 statistical analysis software (SAS Institute, 2021, Cary, NC, USA). Differences were considered significant at a corrected *p*-value < 0.05.

## 5. Conclusions

This study is useful because it investigated the differences in the characteristics between persistent *S. aureus* and CoNS bacteremia in terms of methicillin resistance. Limitations of this study include the small number of patients, especially those with persistent methicillin-susceptible CoNS bacteremia. However, this is the first study to focus on the differences in the clinical features of persistent CoNS bacteremia in patients with and without methicillin resistance, making the report clinically significant. The main conclusions of this study are as follows: (1) the early (30-day), late (30–90 days), and 90-day mortality rates in the persistent CoNS bacteremia group were similar to those in the persistent *S. aureus* bacteremia group; (2) the persistent MRSA bacteremia group tended to have higher early (30-day), late (30–90 days), and 90-day mortality rates than the persistent MSSA bacteremia group, although the difference was not statistically significant; and (3) the persistent MRSA bacteremia non-clearance group tended to have higher early (30-day), late (30–90 days), and 90-day mortality rates than the persistent MRSA bacteremia clearance group. However, the difference was not statistically significant.

Patients who are immunocompromised or in poor general condition develop persistent CoNS bacteremia and do not have source control of distant metastases, or those who develop persistent MRSA bacteremia and do not have clearance of PB, tend to have higher mortality rates. Effective source-control measures for distant metastases in patients with persistent and metastatic CoNS bacteremia and validation of PB clearance in patients with persistent MRSA bacteremia can improve patient survival outcomes.

## Figures and Tables

**Figure 1 antibiotics-12-00454-f001:**
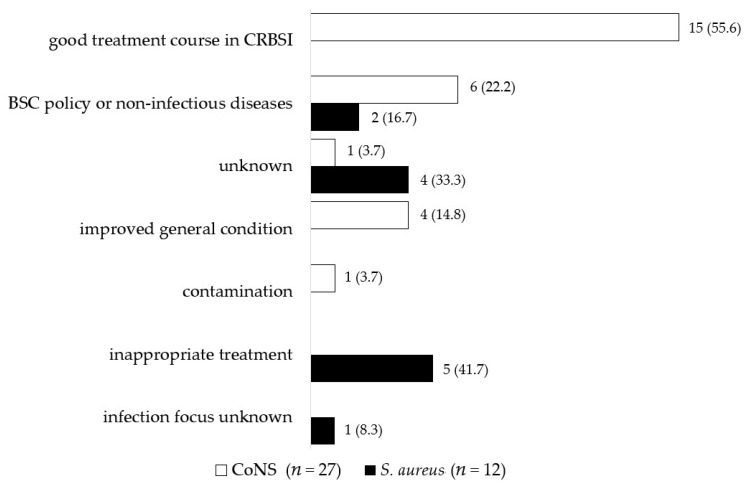
Reasons for non-clearance of persistent staphylococcal bacteremia. Data are presented as numbers (%). A good treatment course for CRBSI was defined as a case of CRBSI with appropriate antimicrobial therapy, good source control, and a good clinical course (e.g., improvement in consciousness, vital signs, and general condition); therefore, negative BCs were not confirmed. The underlying diseases of patients who became part of the BSC policy were gastric cancer, interstitial pneumonia, cerebral infarction, and disseminated intravascular coagulation. Non-infectious diseases are cases in which the clearance of PB cannot be confirmed due to the death of a patient from a disease other than an infectious disease, interstitial pneumonia, or acute pancreatitis. An improved general condition means that negative BC was not confirmed due to improved general condition. Inadequate treatment refers to patients with inadequate duration of antimicrobial therapy, poor choice of regimen, or inadequate source control. Antimicrobial therapy was deemed inappropriate if at least one of the following conditions was met: administration of inefficacious antimicrobial agents, i.e., agents that did not effectively treat infections with organisms identified in the BC; continuation of the initial antimicrobial agents even though the susceptibility test result was known, and de-escalation was possible; and administration of antibiotic therapy for a shorter period than the current medical standards. BC, blood culture; BSC, best supportive care; CoNS, coagulase-negative staphylococci; CRBSI, catheter-related bloodstream infection; *S. aureus*, *Staphylococcus aureus*.

**Table 1 antibiotics-12-00454-t001:** Clinical characteristics of persistent *Staphylococcus aureus* bacteremia and persistent coagulase-negative *Staphylococci bacteremia*.

	Persistent *S. aureus* Bacteremia (*n* = 92)	Persistent CoNS Bacteremia (*n* = 88)	Odds Ratio [95% CI]	*p*-Value
**Demography**				
Sex (male, %)	65 (70.7)	48 (54.5)	2 [1.1, 3.7]	0.031
Age, years, median (IQR)	63.5 (56.5–69.3)	55.5 (40.3–58.8)		0.003
**Comorbidities**				
Diabetes mellitus	25 (27.2)	9 (10.2)	3.3 [1.4, 7.5]	0.004
ESDR on hemodialysis	11 (12.0)	6 (6.8)	1.9 [0.7, 5.3]	
Liver cirrhosis	13 (14.1)	7 (8.0)	1.9 [0.7, 5]	
Solid malignancy	28 (30.4)	24 (27.3)	1.2 [0.6, 2.3]	
Hematologic malignancy	2 (2.2)	15 (17.0)	0.1 [0, 0.5]	0.001
Neutropenia	1 (1.1)	7 (8.0)	0.1 [0, 1.1]	0.032
Immunosuppression	12 (13.0)	24 (27.3)	0.4 [0.2, 0.9]	0.025
**Vital signs**				
BMI, kg/m^2^, median (IQR)	21.1 (18.4–23.8)	22.2 (18.6–24.1)		
Body temperature, °C, median (IQR)	38.7 (38.0–39.1) (*n* = 82)	38.0 (37.5–38.9) (*n* = 84)		0.009
**Laboratory markers**				
White blood cell count, 10^9^/L, median (IQR)	9550.0 (7375.0–12,200.0)	8000.0 (4100.0–12,725.0)		0.022
Neutrophil count, 10^9^/L, median (IQR)	8160.0 (6210.0–10,950.0) (*n* = 89)	5940.0 (3190.0–10,697.5) (*n* = 81)		0.006
C-reactive protein, mg/dL, median (IQR)	10.0 (4.6–16.9)	4.2 (1.7–7.6)		<0.001
**Devices**				
Intravascular device	53 (57.6)	76 (86.4)	0.2 [0.1, 0.4]	<0.001
Intravascular device removal	46 (86.8)	58 (76.3)	2 [0.8, 5.3]	
Cardiovascular surgery	28 (30.4)	14 (15.9)	2.5 [1.2, 5.3]	0.013
ECMO	0 (0)	2 (2.3)	0	
Continuous hemodiafiltration	0 (0)	19 (21.6)	0	
Mechanical ventilation	24 (26.1)	20 (22.7)	1.2 [0.6, 2.4]	
**Status of persistent bacteremia**				
Period until FUBC is carried out, median (IQR)	3.0 (2.0–4.0)	3.0 (1.0–5.0)		
Duration of bacteremia, median (IQR)	3.5 (2.0–6.3)	3.0 (1.0–7.0)		
**Site of infection**				
CRBSI	40 (29.2)	73 (77.7)	0.1 [0.1, 0.2]	<0.001
Abscess	15 (10.9)	1 (1.1)	11.4 [1.5, 88.1]	0.003
Infectious endocarditis	14 (10.2)	2 (2.1)	5.2 [1.2, 23.6]	0.018
Endovascular devices infections	12 (8.8)	1 (1.1)	8.9 [1.1, 69.9]	0.017
Septic embolism	10 (7.3)	1 (1.1)	7.3 [0.9, 58.2]	0.030
Pyogenic spondylitis	10 (7.3)	1 (1.1)	7.3 [0.9, 58.2]	0.030
Thrombophlebitis	9 (6.6)	4 (4.3)	1.6 [0.5, 5.3]	
Surgical site infection	5 (3.6)	0 (0)	–	
Suppurative arthritis	4 (2.9)	0 (0)	–	
Skin and soft tissue infections	3 (2.2)	1 (1.1)	2.1 [0.2, 20.3]	
Osteomyelitis	2 (1.5)	1 (1.1)	1.4 [0.1, 15.4]	
Others	4 (2.9)	0 (0)	–	
Unknown	9 (6.6)	9 (9.6)	0.7 [0.3, 1.7]	
**Hospital stays**				
Duration of hospitalization, days, median (IQR)	59.5 (36.8–104.3)	78.0 (39.5–127.0)		
Presence of ICU	28 (30.4)	40 (45.5)	0.5 [0.3, 1]	0.046
Duration of ICU stay, days, median (IQR)	0 (0–8)	0 (0–18.3)		0.034
**Intervention**				
The use of antibiotics (Appropriate)	76 (82.6)	81 (92.0)	0.4 [0.2, 1.1]	
Source control	61 (66.3)	66 (75.0)	0.7 [0.3, 1.3]	
**Mortality**				
Early (30-day) mortality	5 (5.4)	2 (3.2)	2.5 [0.5, 13.1]	
Late (30–90 days) mortality	2 (2.2)	4 (6.5)	0.5 [0.1, 2.6]	
90-day mortality	7 (7.6)	6 (9.7)	0.9 [0.3, 2.8]	

Data are presented as numbers (%) unless indicated otherwise. *p*-Values are only listed in the table for values that showed significant differences. Blood tests were performed on the same day as blood culture collection. Immunosuppression was considered in the presence of neutropenia, hematopoietic stem cell transplantation, solid organ transplantation, and corticosteroid therapy (prednisone, 16 mg/day for 15 days). Cardiovascular surgery includes valve replacement, vascular graft replacement, ventricular-assisted device implantation, and cardiac-device implantation. Endovascular device infections involve vascular grafts, pacemakers, implantable cardioverter-defibrillators, and left ventricular assist devices. BMI, body mass index; CI, confidence intervals; CoNS, coagulase-negative staphylococci; CRBSI, catheter-related bloodstream infection; ECMO, extracorporeal membrane oxygenation; ESDR, end-stage renal disease; FUBC, follow-up blood culture; ICU, intensive care unit; IQR, interquartile range; *S. aureus*, *Staphylococcus aureus*.

**Table 2 antibiotics-12-00454-t002:** Clinical characteristics of persistent *Staphylococcus aureus* bacteremia and persistent coagulase-negative *Staphylococci bacteremia* in terms of methicillin-resistant status.

	Persistent MRSA Bacteremia (*n* = 43)	Persistent MSSA Bacteremia (*n* = 49)	Odds Ratio [95% CI]	*p*-Value	Persistent MRCoNS Bacteremia (*n* = 70)	Persistent MSCoNS Bacteremia (*n* = 18)	Odds Ratio [95% CI]	*p*-Value
**Demography**								
Sex (male, %)	31 (72.1)	34 (69.4)	1.1 [0.5, 2.8]		37 (52.9)	11 (61.1)	0.7 [0.2, 2.1]	
Age, years, median (IQR)	66.0 (64.3–67.8)	57.0 (51.5–69.3)			58.0 (53.0–59.0)	43.5 (33.3–63.3)		
**Comorbidities**								
Diabetes mellitus	11 (25.6)	14 (28.6)	0.9 [0.3, 2.2]		8 (11.4)	1 (5.6)	2.2 [0.3, 18.8]	
ESDR on hemodialysis	4 (9.3)	7 (14.3)	0.6 [0.2, 2.3]		4 (5.7)	2 (11.1)	0.5 [0.1, 2.9]	
Liver cirrhosis	6 (14.0)	7 (14.3)	1 [0.3, 3.2]		6 (8.6)	1 (5.6)	1.6 [0.2, 14.1]	
Solid malignancy	16 (37.2)	12 (24.5)	1.8 [0.7, 4.5]		19 (27.1)	5 (27.8)	1 [0.3, 3.1]	
Hematologic malignancy	2 (4.7)	0 (0)	–		15 (21.4)	0 (0)	–	0.034
Neutropenia	1 (2.3)	0 (0)	–		7 (10.0)	0 (0)	–	
Immunosuppression	8 (18.6)	4 (8.2)	2.6 [0.7, 9.2]		21 (30.0)	2 (11.1)	3.7 [0.8, 17.3]	
**Devices**								
Intravascular device	25 (58.1)	28 (57.1)	1 [0.5, 2.4]		66 (94.3)	13 (72.2)	3.5 [0.9, 12.6]	
Intravascular device removal	21 (84.0)	25 (89.3)	0.6 [0.1, 3.1]		55 (83.3)	7 (53.8)	4.3 [1.2, 15.2]	0.028
Cardiovascular surgery	16 (37.2)	12 (24.5)	1.8 [0.7, 4.5]		12 (17.1)	3 (16.7)	1.1 [0.3, 4.3]	
ECMO	0 (0)	0 (0)	–		0 (0)	2 (11.1)	0	0.040
Continuous hemodiafiltration	8 (18.6)	7 (14.3)	1.4 [0.5, 4.2]		18 (25.7)	1 (5.6)	5.9 [0.7, 47.4]	
Mechanical ventilation	15 (34.9)	9 (18.4)	2.4 [0.9, 6.2]		17 (24.3)	3 (16.7)	1.6 [0.4, 6.2]	
**Status of persistent** **bacteremia**								
The period until FUBC is carried out, median (IQR)	3.0 (2.0–4.0)	2.0 (2.0–4.0)			3.0 (1.0–5.0)	3.0 (1.0–4.0)		
Duration of bacteremia, median (IQR)	6.0 (3.0–9.0)	3.0 (2.0–4.0)		<0.001	3.0 (1.3–7.0)	3.5 (1.3–7.0)		
**Hospital stays**								
Duration of hospitalization, days, median (IQR)	63.0 (42.5–99.0)	56.0 (36.0–108.0)			93.0 (43.0–138.5)	53.0 (31.0–82.0)		0.029
**Intervention**								
Use of antibiotics (Appropriate)	36 (83.7)	40 (81.6)	1.2 [0.4, 3.4]		65 (92.9)	16 (88.9)	1.6 [0.3, 9.2]	
Source control	29 (67.4)	32 (65.3)	1.1 [0.5, 2.6]		53 (75.7)	13 (72.2)	1.2 [0.4, 3.9]	
**Mortality**								
Early (30-day) mortality	4 (9.3)	1 (2.0)	4.9 [0.5, 45.9]		2 (2.9)	0 (0)	–	
Late (30–90 days) mortality	1 (2.3)	1 (2.0)	1.1 [0.1, 18.8]		3 (4.3)	1 (5.6)	0.8 [0.1, 7.8]	
90-day mortality	5 (11.6)	2 (4.1)	3.1 [0.6, 16.8]		5 (21.4)	1 (5.6)	1.3 [0.1, 11.9]	

Data are presented as numbers (%) unless indicated otherwise. *p*-values are only listed in the table for values that showed significant differences. Blood tests were performed on the same day as the blood culture sample collection. Immunosuppression was considered in the presence of neutropenia, hematopoietic stem cell transplantation, solid organ transplantation, and corticosteroid therapy (prednisone, 16 mg/day for 15 days). Cardiovascular surgery includes valve replacement, vascular graft replacement, ventricular-assisted device implantation, and cardiac device implantation. Endovascular device infections involve vascular grafts, pacemakers, implantable cardioverter-defibrillators, and left ventricular assist devices. CI, confidence intervals; CoNS, coagulase-negative staphylococci; ECMO, extracorporeal membrane oxygenation; ESDR, end-stage renal disease; FUBC, follow-up blood culture; IQR, interquartile range; MRCoNS, methicillin-resistant CoNS; MRSA, methicillin-resistant *S. aureus*; MSCoNS, methicillin-sensitive CoNS; MSSA, methicillin-sensitive *S. aureus*; *S. aureus*, *Staphylococcus aureus*.

## Data Availability

The datasets created and analyzed during the current study are not publicly available because they contain a large amount of detailed patient information. The dataset was obtained from the Department of Infectious Diseases, Internal Medicine, Tohoku University Graduate School.

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
