# Peer review of "Clinical Characteristics and Outcomes of Persistent Staphylococcal Bacteremia in a Tertiary Care Hospital"

_antibiotics, 2023, doi:10.3390/antibiotics12030454_

Round 1

Reviewer 1 Report

Congratulations on the paper and work.

Its a very hard and good analyse, neverless its not an easy reading. Its a lot of data, and could be very good, if you extract more conclusion / discussion.

Keep up good work.

Author Response

Its a very hard and good analyse, neverless its not an easy reading. Its a lot of data, and could be very good, if you extract more conclusion / discussion.

Response: I am deeply appreciative of your incisive observations. Indeed, the sheer volume of data may have rendered it challenging for readers to comprehend. Hence, we have excised items deemed to possess low clinical significance and negligible variation from tables 1 and 2, thereby facilitating their perusal. Additionally, the conclusions and discussion have been revised to be more detailed and concise, taking into account the comments received from the reviewers;

“Furthermore, our hospital provides aggressive treatment for children with acute lymphoblastic leukemia and acute myeloid leukemia, which may have contributed to the persistent CoNS bacteremia group being significantly younger than the persistent S. aureus bacteremia group.” (Lines 206–209)

“Effective source-control measures for distant metastases in patients with persistent and metastatic CoNS bacteremia and validation of PB clearance in patients with persistent MRSA bacteremia can improve patient survival outcomes.” (Lines 408–410)

Reviewer 2 Report

Dear authors 

The writing of the manuscript is good and acceptable. The findings have revealed no significant difference between patients infected with MRSA and MSSA or CoNS strains. The reasons of differences between groups in terms of blood parameters and demographics considering non-significant mortality rates should be clarified. 

The p value scores of the table2 is unclear. 

It is also conceivable that despite higher persistence rate of MRSA versus each of MSSA or CoNS, the mortality is not significantly different. 

If the study is among those patients with hematological malignancies or neutropenia, it should be stated in the title and Abstract. 

A brief methodology about the isolation of MRSA, MSSA and CoNS isolates can improve the study. 

Best regards

Author Response

  1. The reasons of differences between groups in terms of blood parameters and demographics considering non-significant mortality rates should be clarified.

Response: We are grateful for your insightful recommendations regarding our manuscript. We think there is a great need to discuss the relationship between blood parameters, demographics, and mortality rates. Therefore, we conducted additional studies to clarify their relationship, but it was difficult to clarify the causal relationship. However, we have tried to be as responsive as possible to the reviewer's advice by adding in the manuscript the reasons for the significant differences in blood parameters and demographics between the persistent S. aureus and CoNS bacteremia groups;

“Higher number of patients in the persistent CoNS bacteremia group with hematological malignancies and neutropenia may have contributed to the significantly lower white blood cell and neutrophil counts in the persistent CoNS bacteremia group than in the persistent S. aureus bacteremia group.” (Lines 202–205)

“Furthermore, our hospital provides aggressive treatment for children with acute lymphoblastic leukemia and acute myeloid leukemia, which may have contributed to the persistent CoNS bacteremia group being significantly younger than the persistent S. aureus bacteremia group.” (Lines 206–209)

  1. The p value scores of the table2 is unclear.

Response: We express our gratitude to the reviewer for their feedback. To enhance readability, p-values are displayed solely when significant disparities are identified. This has been recorded in the annotations section of table 2;

P-values are only listed in the table for values that showed significant differences.” (Lines 143–144)

  1. It is also conceivable that despite higher persistence rate of MRSA versus each of MSSA or CoNS, the mortality is not significantly different.

Response: We extend our gratitude to the reviewer for the valuable comment. Undoubtedly, the reviewer's assessment is accurate, with a marked prolongation of bacteremia observed in the MRSA cohort compared to the MSSA cohort, yet no substantial disparity in mortality was evident.

However, early and 90-day mortality rates were established to be higher in the MRSA group with odds ratios of 4.9 and 3.1 respectively. A potential explanation for the insufficiency of a considerable difference in mortality may be attributed to the limited sample size. Therefore, it is hypothesized that if future studies feature an increased cohort size, a significant difference may become obvious. With regard to the sample size limitation, we had already noted in the Limitations section that the number of MSCoNS patients is modest, but we have added a nuance regarding the equally small number of patients infected with other bacterial species.

“Limitations of this study include the small number of patients, especially those with persistent methicillin-susceptible CoNS bacteremia.” (Lines 392–394)

  1. If the study is among those patients with hematological malignancies or neutropenia, it should be stated in the title and Abstract.

Response: Thank you for bringing that to our attention. Although a substantial number of patients in this study presented with hematologic malignancies and/or neutropenia, particularly those with persistent CoNS bacteremia, it would be erroneous to assume that all participants had such conditions. Therefore, stating that this study was conducted in patients with hematologic tumors or neutropenia may be misleading to the reader. Hence, in the discussion section, we have specified that neutropenia is frequently encountered during treatment, as our institution routinely cares for individuals with hematologic tumors;

“Our hospital, a university facility with many departments, actively treats patients with hematological malignancies. Because of the administration of immunosuppressive drugs in combined with anticancer therapy and bone marrow transplantation, such patients are frequently immunocompromised, resulting in a presumed high incidence of neutropenia.” (Lines 194–198)

  1. A brief methodology about the isolation of MRSA, MSSA and CoNS isolates can improve the study.

Response: My heartfelt gratitude for reviewer’s astute observation. In consonance with the reviewer's sagacious remark, we firmly believe that articulating the identification procedure of the bacterial species is paramount for the study's reproducibility. Hence, we have incorporated the following elucidation in the materials and methods section;

“VITEK MS System (BioMérieux, Métropole de Lyon, France) was used to identify bacterial species of S. aureus and CoNS, and a Walk Away-96 Plus system (Siemens Healthcare Diagnostics, Deerfield, IL, USA) was used for susceptibility testing.” (Lines 369–372)

Reviewer 3 Report

Authors aimed to clarify the differences in the clinical characteristics of persistent S. aureus and CoNS bacteremia. For this, authors performed a secondary data analysis utilizing electronic clinical charts and hospital records from the Tohoku University Hospital between January 2012 and December 2021. This paper should be of interest for the readers of the Journal. 

> Authors should review the cited references. Recent references are necessary.

Author Response

Authors should review the cited references. Recent references are necessary.

Response: We are grateful for the reviewer’s illuminating remarks. It is of utmost importance that we assimilate the discoveries of contemporary investigations. Hence, we have incorporated references to additional recent research studies (references number 15, 27, 36, and 40).

Reviewer 4 Report

The study topic is good. I have some suggestions for the authors.

1) Mention the place and settings of study in line 314 and 315.

2) Mention the out some more clearly in your study.

3) What is the statement of the problem? Mentioned in your last paragraph of introduction.

4) Table 1 and table 2 have demography better to present small and more clear tables.

5) The data of table 1 and 2 is the same. So, better to present in small tables.

6) Figure 1, good treatment course? on what basis is it good treatment?

7) Line 166, which is antimicrobial therapy, mentioned here.

8) What is your recommendation, mentioned in your conclusion part.

Author Response

  1. Mention the place and settings of study in line 314 and 315.

Response: We are grateful for your insightful recommendations regarding our manuscript. We regard it as essential to mention the place and settings of the study. Thus, we have delineated these specifics at the commencement of the study design and setting, and have also incorporated information into the line designated by the reviewer;

“For secondary data analyses, we used electronic clinical charts and hospital records from Tohoku University Hospital between January 2012 and December 2021.” (Lines 325–326)

“Data for this study were extracted from the computerized records of the Department of Laboratory Medicine and medical records and databases of the Department of Infectious Diseases, Tohoku University Hospital.” (Lines 326–329)

  1. Mention the out some more clearly in your study.

Response: My sincerest thanks for reviewer’s perceptive observation. In accordance with reviewer’s suggestion, we have integrated a succinct portrayal of the outcomes garnered from this study within the conclusion section, along with more granular information regarding therapeutic interventions.

“Patients who are immunocompromised or in poor general condition develop persistent CoNS bacteremia and do not have source control of distant metastases, or those who develop persistent MRSA bacteremia and do not have clearance of PB tend to have higher mortality rates.” (Lines 405–408)

“Effective source-control measures for distant metastases in patients with persistent and metastatic CoNS bacteremia and validation of PB clearance in patients with persistent MRSA bacteremia can improve patient survival outcomes.” (Lines 408–410)

  1. What is the statement of the problem? Mentioned in your last paragraph of introduction.

Response: We extend our gratitude to the reviewer for the valuable comment. We agree with it is important to raise the issue in the introduction section to clarify the purpose of the manuscript. Therefore, in the last paragraph of the introduction, we have described the limitations and problems of the information we know from the current published papers.

“However, few studies have compared the clinical characteristics of PB caused by these species. Additionally, most CoNS studies have primarily focused on PB among infants in the NICUs, and information on persistent CoNS bacteremia in adults is lacking. Furthermore, few studies have compared the outcomes of methicillin resistance in persistent S. aureus and CoNS bacteremia.” (Lines 60–64)

  1. Table 1 and table 2 have demography better to present small and more clear tables.

Response: Grateful for the reviewer’s astute observation. Indeed, as the reviewer astutely noted, the table was convoluted and challenging to peruse, owing to its plethora of items. Hence, in a bid to enhance its readability, we have purged both tables 1 and 2 of items deemed to have negligible disparity and low clinical relevance. As for table 2, we have added a brief note in the manuscript regarding the data that did not show significant differences.

“There were no significant differences between persistent MRSA and methicillin-susceptible S. aureus (MSSA) bacteremia or between persistent MRCoNS and MSCoNS bacteremia regarding vital signs, laboratory markers, site of infection, and the presence or duration of ICU, high care unit, and cardiac care unit stay.” (Lines 111–115)

  1. The data of table 1 and 2 is the same. So, better to present in small tables.

Response: We extend our appreciation for your insightful recommendations for our manuscript. As the reviewer mentioned, tables 1 and 2 are complicated by many similar elements. Therefore, we focused mainly on table 2 and omitted elements that were not clinically important and not significantly different, such as vital signs and laboratory markers.

  1. Figure 1, good treatment course? on what basis is it good treatment?

Response: We extend our gratitude to the reviewer for the valuable suggestion. We thought that clarifying the definition of good treatment course would help readers better understand the content. Therefore, we have added the definition of "good treatment" in the remarks section of figure 1, and good clinical course also includes specific details.

“A good treatment course for CRBSI was defined as a case of CRBSI with appropriate antimicrobial therapy, good source control, and a good clinical course (e.g., improvement in consciousness, vital signs, and general condition)” (Lines 175–177)

  1. Line 166, which is antimicrobial therapy, mentioned here.

Response: We express our gratitude to the reviewer for their feedback. The precise definition of proper use of an antimicrobial therapy is deemed imperative. The definitions of appropriate and inapplicable use of these agents have been delineated in the annotation’s column of figure 1.

“Antimicrobial therapy was deemed inappropriate if at least one of the following conditions was met: administration of inefficacious antimicrobial agents, i.e., agents that did not effectively treat infections with organisms identified in the BC; continuation of the initial antimicrobial agents even though the susceptibility test result was known and de-escalation was possible; and administration of antibiotic therapy for a shorter period than the current medical standards.” (Lines 183–188)

  1. What is your recommendation, mentioned in your conclusion part.

Response: Thank you for bringing that to our attention. We consider it essential to incorporate specific recommendations rooted in the considerations gleaned from the findings of this study. As such, we have integrated specific recommendations concerning diagnosis and therapy in the section of conclusions.

“Effective source-control measures for distant metastases in patients with persistent and metastatic CoNS bacteremia and validation of PB clearance in patients with persistent MRSA bacteremia can improve patient survival outcomes.” (Lines 408–410)

Round 2

Reviewer 4 Report

After modification, The paper seems good in all aspects. Author incorporated the changing accordingly in the paper, Hence paper may be accepted.